# Presence of HPV with overexpression of p16INK4a protein and EBV infection in penile cancer—A series of cases from Brazil Amazon

Valquíria do Carmo Alves Martins[1,2,3]*, Isabela Werneck Cunha[4,5], Giuseppe Figliuolo[1,6], Heidy Halanna de Melo Farah Rondon[2], Paloma Menezes de Souza[6], Felipe Luz Torres Silva[2], Guilherme Luz Torres Silva[6], Michele de Souza Bastos[7], Daniel Barros de Castro[3,8], Monique Freire Santana[1,7], Rajendranath Ramasawmy[3,7,9], José Eduardo Levi[10], Kátia Luz Torres[1,2,3]

1 Department of Education and Research, Fundação Centro de Controle de Oncologia do Estado do Amazonas, Manaus, Amazonas, Brazil, 2 Universidade Federal do Amazonas, Manaus, Amazonas, Brazil, 3 Genomic Health Surveillance Network: Optimization of Assistance and Research in The State of Amazonas – REGESAM, Manaus, Amazonas, Brazil, 4 Department of Pathology, AC Camargo Cancer Center, São Paulo, Brazil, 5 Department of Pathology, Rede D'OR- São Luiz, São Paulo, Brazil, 6 Universidade do Estado do Amazonas, Manaus, Amazonas, Brazil, 7 Department of Virology, Fundação de Medicina Tropical Heitor Vieira Dourado, Manaus, Amazonas, Brazil, 8 Technical Advisory, Fundação de Vigilância em Saúde do Amazonas, Manaus, Amazonas, Brazil, 9 Faculdade de Medicina, Universidade Nilton Lins, Manaus, Amazonas, Brazil, 10 Instituto de Medicina Tropical, Universidade de São Paulo, São Paulo, Brazil

☯ These authors contributed equally to this work.
* alvesvalquiria@yahoo.com.br

## Abstract

### Background

In Brazil, penile cancer (PC) is not uncommon. The highest incidence of PC is in the North and Northeast of the country. In addition to phimosis, the Human Papillomavirus (HPV) and Epstein-Baar Virus (EBV) infections are also related as risk factors for PC. The overexpression of p16INK4a is a surrogate sensitive marker of HPV infection in PC.

### Objectives

To correlate p16INK4a overexpression and HPV infection status with EBV infection in a series of PC patients from the Amazon region.

### Methods

Tumor tissues from 47 PC cases were analyzed for the presence of HPV and EBV DNA by PCR. All PC patients were diagnosed between 2013 and 2018 at a public reference cancer center hospital in Manaus, Amazonas—Brazil. HPV was genotyped using E7 HPV16/ HPV18 type-specific real-time PCR and the PapilloCheck® HPV-Screening assay. p16INK4a expression was evaluated by immunohistochemistry using the automated Ventana® Bench-Mark Ultra.

**Data Availability Statement:** All relevant data are in the paper and in the Supporting information.

**Funding:** This work was supported by The Fundação de Amparo à Pesquisa do Estado do Amazonas- FAPEAM for financial support (PROGRAMA UNIVERSAL AMAZONAS N.021/2011-FAPEAM; PROGRAMA PAPAC N.020/2013 – FAPEAM; PROGRAMA PAPAC N.05/2019 – FAPEAM and PROGRAMA PRó ESTADO N.002./2008-FAPEAM). The Rede D'OR- São Luiz provided support in the form of a salary for author Isabela Werneck Cunha IW, but did not have any additional role in the study design, data collection and analysis, decision to publish, or preparation of the manuscript. The specific roles of this author is articulated in the 'author contributions' section. The Genomic Health Surveillance Network: Optimization of Assistance and Research in the State of Amazonas – REGESAM is not a funder but a network of researchers in the Genomic field on the State of Amazonas.

**Competing interests:** The authors have read the journal's policy and have the following conflicts: author Isabela Werneck Cunha IW is affiliated with The Rede D'OR- São Luiz. This does not alter our adherence to all the PLOS ONE policies on sharing data and materials.

## Results

The mean age of patients at the time of diagnosis was 57.4 years ±SD 17.8 ranging from 20 to 90 years old. Most of the patients (64%) came from rural areas of the Amazonas State. Thirty patients had phimosis (64%). Among the patients with phimosis, 43% (13/30) underwent circumcision, three during childhood and 10 in adulthood. 60% of the patients were smokers or ex-smokers. HPV infection was observed in 45% (21/47) of cases. HPV16 was detected in 13 patients (61%). Other HPV types detected were HPV 6, 11, 42, 51, 53, 68 and 44/55. EBV infection was observed in 30% (14/47) of the patients with PC. Co-infection with HPV and EBV was observed in 28% (6/21) cases. p16INK4a was only investigated in 26 samples. The p16INK4a overexpression was observed exclusively in HPV 16 positive cases and four HPV negative cases. In the survival analysis, the follow-up time was 35.4 months/patient. The mortality rate during the follow up time was 38%.

## Conclusions

p16INK4a positivity presented a high correlation to HPV 16 DNA detection, reinforcing its use as a surrogate marker for HPV-driven cancers. Infection with EBV was quite frequent and its role in epithelial penile oncogenesis needs to be demonstrated.

## Introduction

The occurrence of Penile Cancer (PC) varies worldwide. In developed countries, PC has a low incidence, corresponding to 0.3–1% of malignant neoplasms in men. In some developing countries, the incidence of PC may be much higher than the global average incidence [1–3]. In Brazil, PC accounts for approximately 2.1% of all tumors in men being the highest incidence reported in the Latin America (2.9–6.8 cases per 100,000 men-years) [3]. We can find regional rate differences along the country. In the North and Northeast of Brazil, PC incidence is five times higher compared to the Midwest, South and Southeast regions [4]. According to the Brazilian National Cancer Institute (INCA), the death rate from PC in the northern region of Brazil has doubled in the last decade, from 0.05% to 0.10% [5].

PC is a multifactorial disease and the risk factors and/or favorable conditions to develop PC are not fully established. Phimosis and smegma accumulation are observed in more than 80% of the patients with PC associated with chronic inflammation process [6,7]. Other factors, such as smoking and sexually transmitted infection, are also related to the onset of neoplasms [7,8]. HPV infection is present in approximately 50% of PC cases and the most prevalent genotype is HPV16 (30%) [7,9–11]. A recent meta-analysis study of 52 studies showed a pooled prevalence of 50.8% (44.8–56.7) of HPV infection in PC with a rate of 68.3% (58.9–77.1) of HPV16 [12].

The role played by HPV in carcinogenesis of the penis appears to be similar to cervical cancer. HPV encodes the E6 and E7 oncogenes which are required for malignant transformation and maintenance of host cells. The viral oncoproteins (E6 and E7) may compromise the regulation of the host cell cycle and lead to an uncontrolled proliferation [13,14]. P16 is a tumor suppressor gene and its protein is physiologically expressed in normal tissues. The inactivation of the retinoblastoma gene (pRb) by HPV E7 results in overexpression of p16INK4a due to the lack of negative feedback loop between pRb and p16 protein [15]. The overexpression of p16INK4a in tumor cells has been shown to correlate with high-risk HPV DNA detection in PC [16].

Epstein-Barr virus (EBV) is another agent associated with PC [17,18]. Inappropriate expression of its latent genes (Latent Membrane Protein) LMP-1, LMP-2A e LMP-2B, involved in cell persistence, may contribute to the development of tumors [19]. EBV is suggested as a viral cofactor rather than a primary carcinogen in Burkitt's lymphoma, Hodgkin's disease, nasopharyngeal carcinoma [17,20]. In HPV-associated cancer, the presence of EBV may also act as a viral cofactor [21]. Several studies have shown the presence of EBV and co-infection with HPV in PC but a relationship between PC and EBV is yet to be established [17,18].

In light of the possible roles of HPV and EBV infection in the development of PC, this study investigated p16INK4a expression and HPV and EBV infection in a series of patients with PC from the Brazilian Amazon region.

## Materials and methods

### Enrollment

A total of 47 patients with PC and no concomitant urological neoplastic diseases participating in the study were attended at the public reference cancer center hospital—Fundação Centro de Controle de Oncologia do Estado do Amazonas/FCECON from 2013 to 2018. The patients were followed at the Urology clinic of the hospital. All patients were surgically treated by total or partial penectomy.

All patients provided written informed consent and were interviewed to fill a questionnaire concerning sociodemographic data and risk factors. Histopathology characteristics of the tumors were obtained from the medical charts.

This study was approved by the internal review board of the Ethics Committee of the FCECON—approval document #2.230.007, August 21, 2017. SISGEN- A5F36C5.

### Biological samples collection

At the moment of surgery, three to five mm$^3$ of tissue fragments (mass of 50-150mg) from the tumor were collected and stored in a dry plastic microtube free of DNAse and RNAse. Samples were stored at -30 ˚C until processed.

### DNA extraction

DNA was extracted from the frozen tissue (mass of 20-40mg from different parts of the tumor) using the DNeasy® Blood & Tissue Kit (QIAGEN Inc., USA), according to the manufacturer's recommendations. The DNA was eluted in a volume of 200 μL UltraPure™ DNase/RNase-Free Distilled Water (Invitrogen Life Technologies, São Paulo, Brazil).

### Human β-globin PCR

For DNA extraction quality control, the human β-globin gene was amplified by PCR as previously described in the literature with the following pair of primers: GH20: (5'GAAGAGC CAAGGACAGGTAC'3) and PCO4: (5'CAACTTCATCCACGTTCACC'3) generating a 270 bp DNA fragment [22].

### HPV detection

All samples were submitted to generic HPV PCR using the consensus primers (PGMY09/11) which amplifies a 450 bp DNA fragment within the L1 region of mucosal HPVs [23]. Amplification was carried out as previously described using 50–100 ng of DNA in 25 μL of reaction mixture and a thermocycling profile of 1 cycle at 5 min at 95 ˚C, followed by 40 cycles: 1 min at 95 ˚C, 1 min at 55˚C, and 1 min at 72 ˚C, with a final extension for 10 min at 72 ˚C. The

PCR products (450 bp DNA) were analyzed on 1.5% agarose gel stained with SYBR™ Safe DNA Gel Stain (Invitrogen Life Technologies, São Paulo, Brazil) for visualization of DNA under UV light and 100 bp DNA ladder was used as molecular weight control pattern. Precautions to avoid contamination were followed. DNA from the HeLa cell line which harbors 10–20 copies of integrated HPV 18 per cell was used as a positive control in all reactions.

**E7 HPV16/HPV18 type-specific real-time PCR.** All samples were also submitted to two specific TaqMan based real-time qPCR assays targeting either HPV16/HPV18 E7 gene in an ABI 7300 Real-Time PCR System (Applied Biosystems, Foster City, CA). All samples and controls were run in duplicate.

**HPV16-E7.** qPCR assay included the following primers: forward (5'GATGAAATA GATGGTCCAGC3') and reverse (5'GCTTTGTACGCACAACCGAAGC3') primers, and the probe (5'FAM-CAAGCAGAACCGGACAG-MGB-NFQ) in a final reaction volume of 25 μL [24]. Each qPCR reaction contained 1X TaqMan master mix (Applied Biosystems, Foster City, CA), 400 nM each of the forward and reverse primers, 200 nM of fluorogenic TaqMan probe, and 50–100 ng of DNA. The amplification conditions consisted of 50 ˚C for 2 min and 95˚C for 10 min, followed by 40 cycles of 95 ˚C for 15 sec, 55 ˚C for 1 min, and 60 ˚C for 1 minute. DNA from a SiHa cell line which contains 1–2 copies of integrated HPV 16 per cell was used as a positive control in all reactions.

**HPV18-E7.** qPCR assay included the following primers: forward (5'AAGAAAACGAT GAAATAGATGGA3') and reverse (5'GGCTTCCACCTTACAACACA3') primers, and a probe (5'VIC-AATCATCAACATTTACCAGCC-MGBNFQ3') in a final reaction volume of 25 μL, each qPCR contained 1X TaqMan master mix (Applied Biosystems, Foster City, CA), 400 nM each of the forward and reverse primers, 400 nM fluorogenic TaqMan probe, and 50–100 ng of DNA [24]. The amplification conditions consisted of 50˚C for 2 min and 95˚C for 10 min, followed by 40 cycles of 95˚C for 15 sec, 50˚C for 1 min, and 60˚C for 1 min. DNA from the HeLa cell line which harbors 10–20 copies of integrated HPV 18 per cell was used as a positive control in all reactions.

**HPV genotyping—PapilloCheck® HPV-Screening.** All samples that were positive in generic HPV DNA (PGMY09/11) and negative for 16/18 genotypes were submitted to HPV-Screening Test (Greiner Bio-One GmbH, Frickenhausen, Germany) to identify other genotypes. This is a PCR-based DNA microarray system for detection and identification of 24 HPV genotypes, including 16 high-risk HPV genotypes (HPV 16, 18, 31, 33, 35, 39, 45, 51, 52, 56,58, 59, 68, 70, 73, 82), 2 probable high-risk HPV genotypes (HPV 53,66) and 6 low-risk-HPV genotypes (HPV 6, 11, 40, 42, 43, 44/55) [25].

## EBV detection

EBV DNA detection was performed as described elsewhere [26]. Briefly, a sensitive multiplex PCR which amplifies 182 bp within the Exons 4/5 from the terminal protein RNA of EBV and a fragment of human β-actin 450 bp as internal control. The following pairs of primers for PCR amplification were used: EP5-AACATTGGCAGCAGGTAAGC and EM3 -ACTTAC CAAGTGTCCATAGGAGC for EBV and B-ACT F–TCTACAATGAGCTGCGTGTG and B-ACT R -CATCTCTTGCTCGAAGTC for β-actin. PCR was performed as follows: 1 cycle at 5 min at 95˚C, followed by 10 cycles of 30 sec at 95˚C, 60 sec at 63˚C and subsequently by 30 cycles of 30 sec at 95˚C, 30 sec at 60˚C, and 30 sec at 72˚C, with a final extension for 40 sec at 72˚C. PCR products were analyzed on 1.8% agarose gel stained with SYBR™ Safe DNA Gel Stain (Invitrogen Life Technologies, São Paulo, Brazil) for visualization of DNA under UV light and 100 bp DNA ladder was used as molecular weight control pattern. An EBV positive known sample from the laboratory was used as control.

## Immunohistochemistry for p16$^{INK4a}$

Twenty six samples from PC patients, spotted in tissue microarray were submitted to the immunohistochemistry (IHC) assay for qualitative detection of the p16$^{INK4a}$ (21 samples were not available due to pre-analytical factors). IHC assay was performed in an automated system using the Ventana® BenchMark Ultra according to the manufacturer's instructions. The IHC slides were analyzed by a pathologist. Positivity for p16$^{INK4a}$ was defined as unequivocally nuclear and cytoplasmic staining of at least 70% of the tumor cells [27,28].

## Statistical analysis

Data were compiled using Epi Info™ version: 7.2.2.2 and analyzed using Stata application (v.13, StataCorp, 2013, College Station, Texas, USA). Descriptive analyses were performed using frequency tables. In this study, we analyze independently the relationship between the sociodemographic and clinical profile of patients with PC and three different outcomes: (i) p16$^{INK4a}$ overexpression, (ii) HPV infection status and (iii) EBV infection status. Univariate logistic regression models were used to analyze the relationship between these outcomes and sociodemographic and clinical profile of patients with PC and calculate the odds ratios. A $p$-value <0.05 was considered statistically significant.

Those variables that presented an association at a level of significance of 0.2 in the univariate logistic regression were selected to perform the multiple logistic regression. Multiple logistic regression models were used to identify the independent relationships between clinical characteristics of the patients and the studied outcomes. In this manner, adjusted odds ratios were calculated. A $p$-value <0.05 was considered statistically significant.

Kaplan-Meier method was used to determine the differences between survival time of patients stratified by HPV and EBV infection, and with p16$^{INK4a}$ overexpression. Groups were compared using the log rank test, taking a $p$-value less than 0.05 as statistically significant.

## Results

HPV and EBV were investigated in tumor tissues from 47 patients with PC, aged 20 to 90 years old (mean 57.4 years ±SD 17.8). Ten patients were below the age of 39 (21%). 64% of the patients came from rural areas of the Amazonas State, 23% from Manaus, the capital city and 13% from other Northern states of Brazil. 79% of the patients with PC were Amerindians descendent (mestizo) and 83% had less than eight years of schooling. 32% of the patients had total penectomy and 68% had partial penectomy. Lymphadenectomy was performed in 13/47 (28%) patients (Table 1).

All cases were diagnosed as squamous cell carcinoma. Of the 47 patients with PC, we were able to review and reclassify only 20 patients according to the 2016 WHO classification [29]. The most common histology subtype was "usual" in 11/20 (55%). We could obtain tumor characteristic from the medical records for only 43 patients. 58% of tumors were located in the glans and foreskin and the predominant pattern of growth was verruciform (48%). Other clinical and pathological parameters can be appreciated in Table 1.

In this study, 36% of the patient had a history of cancer in the family. 60% of the patients were smokers or ex-smokers and 64% had phimosis. 43% were circumcised, mostly in adulthood. 28% of the patients reported a history of sexually transmitted diseases once or several times in their adulthood (Table 2).

The prevalence of HPV is shown in Table 3. HPV DNA was detected in 45% (21/47) of the patients with PC. HPV 16 was the most prevalent genotype 61% (13/21). The other 8 genotypes detected were HPV 6, 11, 42, 51, 53, 68 and 44/55. One patient had multiple co-infections; genotypes 16, 42, 44/55. The distribution of EBV and HPV among the patients with PC is

**Table 1. Sociodemographic characteristics of patients diagnosed with penile cancer at Amazon—Brazil.**

| Variables | n | % |
|---|---|---|
| **Age at diagnosis** | | |
| **Mean age** 57,4 (SD 17.8) **(n = 47)** | | |
| 18–39 | 10 | 21 |
| 40–59 | 15 | 32 |
| > 60 | 22 | 47 |
| **Ethnicity (n = 47)**[*] | | |
| Caucasians (whites) | 6 | 13 |
| Blacks | 2 | 4 |
| Mestizo | 37 | 79 |
| Indigenous | 2 | 4 |
| **Formal education time (n = 47)** | | |
| <1 year | 10 | 21 |
| 1–8 years | 29 | 62 |
| 9–12 years | 8 | 17 |
| >12 years | 0 | 0 |
| **Marital status (n = 47)** | | |
| Single | 9 | 19 |
| Married | 26 | 56 |
| Widower | 10 | 21 |
| Divorced | 2 | 4 |
| **Origin (n = 47)** | | |
| Capital city (Manaus) | 11 | 23 |
| Amazonas (interior) | 30 | 64 |
| Other Northern states | 6 | 13 |
| **Penectomy type (n = 47)** | | |
| Partial | 32 | 68 |
| Total | 15 | 32 |
| **Tumor location (n = 43)** | | |
| Glans | 7 | 16 |
| Foreskin | 1 | 2 |
| Glans and Foreskin | 25 | 58 |
| Glans, Foreskin and base | 7 | 17 |
| All over the organ | 3 | 7 |
| **Predominant gross finding (n = 40)** | | |
| Ulcerated | 15 | 37 |
| Verruciform | 19 | 48 |
| Ulcerated and Verruciform | 6 | 15 |
| **Tumor subtype (n = 20)** | | |
| Basaloid | 5 | 25 |
| Warty | 2 | 10 |
| Cuniculatum | 1 | 5 |
| Sarcomatoid | 1 | 5 |
| Usual | 11 | 55 |
| **TNM [30] (n = 32)** | | |
| pTx | 1 | 3 |
| pT1 –pT2 | 18 | 56 |
| pT3- pT4 | 13 | 41 |

(*Continued*)

**Table 1.** (Continued)

| Variables | n | % |
|---|---|---|
| **Histological grade (n = 33)** | | |
| Grade I | 4 | 12 |
| Grade II | 19 | 58 |
| Grade III | 10 | 30 |
| **Lymphadenectomy (n = 47)** | | |
| Yes | 13 | 28 |
| No | 34 | 72 |
| **Metastasis (n = 40)** | | |
| Yes | 12 | 29 |
| No | 28 | 71 |
| **Follow up (n = 47)** | | |
| Dead | 18 | 38 |
| Alive | 29 | 62 |

*self-declared; n: Absolute frequency; %: Relative frequency

shown in Fig 1. EBV infection was observed in 14 out of 47 patients (30%). Co-infection HPV/ EBV was detected in 6 patients, four with HPV 16/EBV and two with HPV 6/EBV and HPV 53/EBV, each. The p16^INK4a overexpression was observed in three patients with co-infection HPV/EBV, all HPV 16.

Association of p16^INK4a overexpression with clinical factors and HPV infection status are shown in Table 4. Overexpression of p16^INK4a was found in 12 cases 46% (12/26). Patients

**Table 2.** Risk factors of the 47 patients diagnosed with penile cancer at Amazon—Brazil.

| Risk factors | n | % |
|---|---|---|
| **Family history of cancer (n = 47)** | | |
| Yes | 17 | 36 |
| No | 30 | 64 |
| **Smoking History (n = 47)** | | |
| Smoking / Ex-Smoking | 28 | 60 |
| No Smoking | 19 | 40 |
| **Phimosis (n = 47)** | | |
| Yes | 30 | 64 |
| No | 17 | 36 |
| **Postectomy (n = 30)** | | |
| Yes | 13 | 43 |
| No | 17 | 57 |
| **Time of postectomy (n = 13)** | | |
| Childhood | 3 | 23 |
| Adulthood | 10 | 77 |
| **Sexually transmitted diseases (n = 47)** | | |
| Yes | 13 | 28 |
| No | 34 | 72 |

n: Absolute frequency; %: Relative frequency

**Table 3. HPV infection prevalence and genotyping at 47 patients diagnosed with penile cancer at Amazon—Brazil.**

| Virus infection (n = 47) | n | % |
|---|---|---|
| HPV—Positive | 21 | 45 |
| HPV—Negative | 26 | 55 |
| **HPV genotyping (N = 21)** | **n** | **%** |
| 6 | 1 | 5 |
| 11 | 1 | 5 |
| 16 | 13 | 61 |
| 44/55 | 1 | 5 |
| 51 | 2 | 9 |
| 53 | 1 | 5 |
| 68 | 1 | 5 |
| 16,42,44/55 | 1 | 5 |

n: Absolute frequency; %: Relative frequency

with phimosis had 11 times more chance of having overexpression of p16$^{INK4a}$ [OR = 11 (95% CI 1.1–109.7); $p$ = 0.04]. The p16$^{INK4a}$ overexpression was observed in the eight HPV 16 positive cases and in four HPV$^-$. Other clinical factors were not significantly associated with HPV infection and p16$^{INK4a}$ overexpression. HPV infection and p16$^{INK4a}$ overexpression were also related to some histological subtypes, HPV infection being positive in the basaloid subtypes (45%) and negative in usual subtypes (81%) (Table 4). Regarding p16$^{INK4a}$ overexpression, of the 11 nonsmokers patients, eight presented over expression of p16$^{INK4a}$ while only four patients of the 15 smokers were with p16$^{INK4a}$ overexpression [OR = 0.13 (95%CI 0.02–0.780); $p$ = 0.026]. Notably, of the eight nonsmoker's patients with p16$^{INK4a}$ overexpression, six had

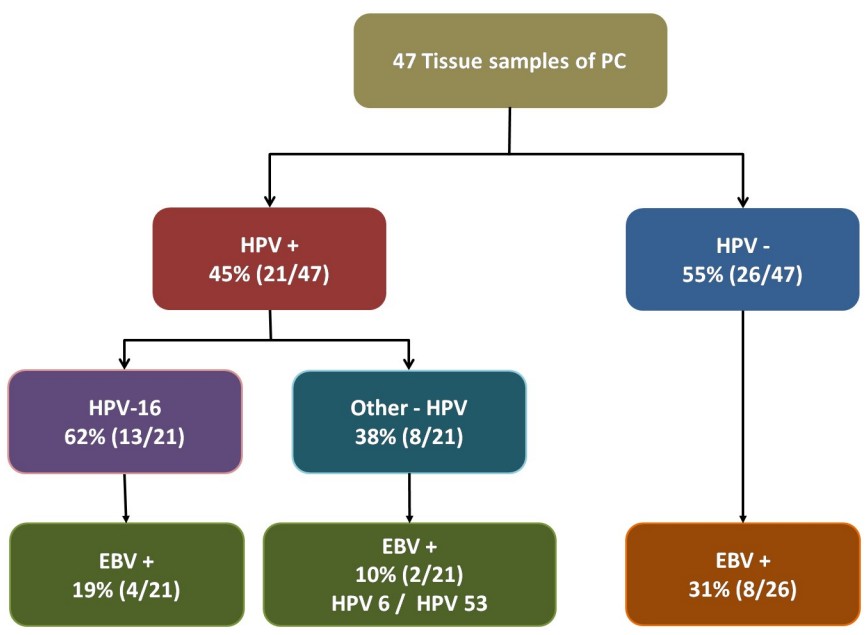

**Fig 1. Distribution of HPV and EBV status among the patients with penile cancer at Amazon—Brazil.** EBV: Epstein-Barr virus; HPV: Human papillomavirus.

**Table 4. Relationship between p16INK4a overexpression, HPV infection status, EBV infection status and clinical factors at 47 patients diagnosed with penile cancer.**

| Variable | p16 | | | | | | HPV | | | | | | EBV | | | | | |
|---|---|---|---|---|---|---|---|---|---|---|---|---|---|---|---|---|---|---|
| | Total | Absent n | Present n | OR | (95% CI) | P value | Total | Absent n | Present n | OR | (95% CI) | P value | Total | Absent n | Present n | OR | (95% CI) | P value |
| **Age (years)** | | | | | | | | | | | | | | | | | | |
| ≤ 45 | 8 | 5 | 3 | | | | 13 | 8 | 5 | | | | 13 | 9 | 4 | | | |
| > 45 | 18 | 9 | 9 | 0.36 | (0.05–2.34) | 0.284 | 34 | 18 | 16 | 1.40 | (0.38–5.24) | 0.597 | 34 | 24 | 10 | 0.94 | (0.23–3.76) | 0.927 |
| **Smoking or Ex Smoking** | | | | | | | | | | | | | | | | | | |
| No | 11 | 3 | 8 | | | | 19 | 11 | 8 | | | | 19 | 13 | 6 | | | |
| Yes | 15 | 11 | 4 | 0.13 | (0.02–0.78) | 0.026* | 28 | 15 | 13 | 1.19 | (0.37–3.86) | 0.770 | 28 | 20 | 8 | 0.87 | (0.24–3.08) | 0.825 |
| **Phimoses** | | | | | | | | | | | | | | | | | | |
| No | 8 | 7 | 1 | | | | 17 | 10 | 7 | | | | 17 | 11 | 6 | | | |
| Yes | 18 | 7 | 11 | 11.0 | (1.10–109.67) | 0.041* | 30 | 16 | 14 | 1.25 | (0.38–4.16) | 0.716 | 30 | 22 | 8 | 0.67 | (0.18–2.40) | 0.535 |
| **Postectomy** | | | | | | | | | | | | | | | | | | |
| No | 11 | 5 | 6 | | | | 17 | 9 | 8 | | | | 17 | 13 | 4 | | | |
| Yes | 7 | 2 | 5 | 2.08 | (0.27–15.77) | 0.477 | 13 | 7 | 6 | 0.96 | (0.23–4.10) | 0.961 | 13 | 9 | 4 | 1.44 | (0.28–7.34) | 0.658 |
| **Histological grading** | | | | | | | | | | | | | | | | | | |
| Grade I / II | 10 | 4 | 6 | | | | 10 | 2 | 8 | | | | 10 | 6 | 4 | | | |
| Grade III/IV | 12 | 6 | 6 | 0.66 | (0.12–3.63) | 0.640 | 23 | 18 | 5 | 0.07 | (0.01–0.46) | 0.005* | 23 | 15 | 8 | 0.80 | (0.17–3.68) | 0.775 |
| **Tumor subtype (n = 20)** | | | | | | | | | | | | | | | | | | |
| Basaloid | 5 | 0 | 5 | | | | 5 | 0 | 5 | | | | 5 | 4 | 1 | | | |
| Warty | 2 | 1 | 1 | | | | 2 | 0 | 2 | | | | 2 | 1 | 1 | | | |
| Cuniculatum | 1 | 1 | 0 | | | | 1 | 0 | 1 | | | | 1 | 0 | 1 | | | |
| Sarcomatoid | 1 | 0 | 1 | | | | 1 | 0 | 1 | | | | 1 | 1 | 0 | | | |
| Usual | 10 | 7 | 3 | | | 0.062 | 11 | 9 | 2 | | | 0.009* | 11 | 7 | 4 | | | 0.540 |
| **TNM (AJCC, 8° ed.)** | | | | | | | | | | | | | | | | | | |
| T1 –T2 | 10 | 5 | 5 | | | | 19 | 10 | 9 | | | | 19 | 13 | 6 | | | |
| T3 –T4 | 11 | 8 | 3 | 0.37 | (0.06–2.30) | 0.290 | 13 | 7 | 6 | 0.95 | (0.23–3.91) | 0,946 | 13 | 9 | 4 | 0.96 | (0.20–4.42) | 0.961 |
| **Lymphadenectomy** | | | | | | | | | | | | | | | | | | |
| No | 16 | 8 | 8 | | | | 34 | 17 | 17 | | | | 34 | 25 | 9 | | | |
| Yes | 10 | 6 | 4 | 0.66 | (0.13–3.30) | 0.619 | 13 | 9 | 4 | 0.44 | (0.11–1.72) | 0.190 | 13 | 8 | 5 | 1.73 | (0.44–6.71) | 0.424 |
| **Metastase** | | | | | | | | | | | | | | | | | | |
| No | 16 | 7 | 9 | | | | 29 | 15 | 14 | | | | 29 | 20 | 9 | | | |
| Yes | 9 | 6 | 3 | 0.38 | (0.07–2.13) | 0.250 | 12 | 7 | 5 | 0.71 | (0.19–2.97) | 0.700 | 12 | 8 | 4 | 1.11 | (0.26–4.66) | 0.886 |
| **Death** | | | | | | | | | | | | | | | | | | |
| No | 12 | 7 | 5 | | | | 29 | 17 | 12 | | | | 29 | 22 | 7 | | | |
| Yes | 14 | 7 | 7 | 0.71 | (0.15–3.38) | 0.671 | 18 | 9 | 9 | 1.41 | (0.43–4.62) | 0.564 | 18 | 11 | 7 | 2.00 | (0.55–7.14) | 0.286 |
| **EBV** | | | | | | | | | | | | | | | | | | |
| Negative | 17 | 9 | 8 | | | | 33 | 18 | 15 | | | | - | - | - | | | |
| Positive | 9 | 5 | 4 | 0.9 | (0.17–4.56) | 0.899 | 14 | 8 | 6 | 0.90 | (0.26–3.17) | 0.870 | - | - | - | - | - | - |
| **HPV** | | | | | | | | | | | | | | | | | | |
| Negative | 11 | 7 | 4 | | | | - | - | - | | | | 26 | 18 | 8 | | | |
| Positive | 15 | 7 | 8 | 2.0 | (0.40–9.83) | 0.394 | - | - | - | - | - | - | 21 | 15 | 6 | 0.90 | (0.25–3.17) | 0.870 |
| **HPV Genotype** | | | | | | | | | | | | | | | | | | |
| HR-HPV | 11 | 3 | 8 | | | | - | - | - | | | | 17 | 12 | 5 | | | |
| LR-HPV | 4 | 4 | 0 | - | - | - | - | - | - | - | - | - | 4 | 3 | 1 | 0.80 | (0.06–9.66) | 0.861 |

HPV: Human papillomavirus; HR-HPV: High-oncogenic risk; LR-HPV: Low oncogenic risk; EBV: Epstein-Barr virus; n: Absolute frequency; Odds Ratio (OR) were calculated by logistic regression.

* statistically significant (p-value < 0.05) and (p-value adjusted < 0.05)

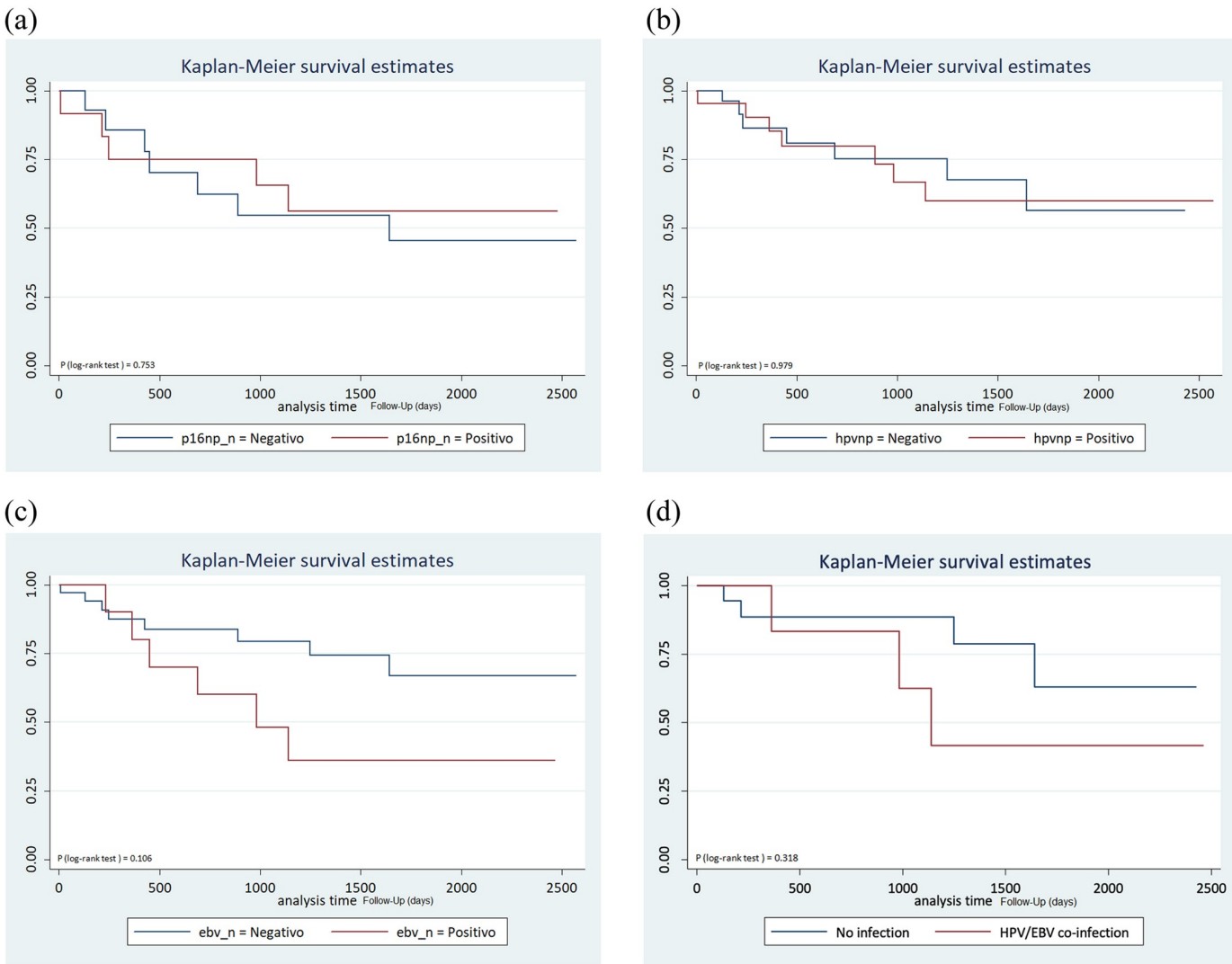

**Fig 2. The y-axis represents the survival function and x-axis represents the follow-up length in days.** Overall survival depending on p16<sup>INK4a</sup> expression (A), HPV infection status (B), positive and negative for EBV (C) and HPV/EBV co-infection (D). EBV: Epstein-Barr virus; HPV: Human papillomavirus.

HPV 16 genotypes and the remaining two were negative for HPV. Among the four smoker's patients with p16$^{INK4a}$ overexpression, two were HPV16 and two negative for HPV. HPV$^+$ patients with PC had poorly differentiated carcinomas (Grade III) compared to HPV$^-$ patients with PC [OR = 0.07 95%CI 0.01–0.047; $p$ = 0.005].

In the survival analysis, the follow-up time was 35.4 months/patient. The mortality rate during the study period was 38% (18/47). Survival analysis was performed by stratification of the patients into p16$^{INK4a+}$ vs p16$^{INK4a-}$, HPV$^+$ vs HPV$^-$, EBV$^+$ vs EBV- and HPV/EBV co-infection vs no infection. Infection status of the deceased and survival patients is shown in S1 Table. There is no evidence of differences in survival of patients according to p16 overexpression (P$_{log\ rank}$ = 0.753), HPV infection (P$_{log\ rank}$ = 0.979), EBV infection (P$_{log\ rank}$ = 0.106) and HPV/EBV co-infection (P$_{log\ rank}$ = 0.318) (Fig 2).

1. **P16 expression and overall mortality**

2. **HPV status and overall mortality**

3. **EBV status and overall mortality**

4. **HPV/EBV co-infection vs no infection and overall mortality**

## Discussion

In the Northern region of Brazil, the mortality rate from PC has doubled in recent years according to INCA [5]. The state of Amazonas, the largest state in Brazil, covers a geographical area of 1,559,168.12 km$^2$ and has about 4 million inhabitants. Half of the population lives in the capital city, Manaus while the other half is distributed irregularly in the rural regions with low density population with very poor access to health services [31].

The disease affects mainly men between the fifth and seventh decade of life [2,6,32]. In this study, the mean age of the diagnosis of PC was 57.4 years, with a higher prevalence in the age group below 59 years. However, young adults less than 30 years were also diagnosed with PC (9%). Other studies in the country have also reported the early occurrence of PC [4,33–37]. The risk factors and carcinogenesis of PC among young adults is still not established. In this study, young patients were at an advanced stage of the disease. Half of them underwent total penectomy and had metastasis. Of the four young patients, one had co-infection HPV 6/EBV and one HPV 44/55. Two were HPV negative. Interestingly, none of them had high-risk HPV. This may suggest that there are other factors or genetic and molecular changes possibly involved in the development of PC in young adults.

In this study, most of the patients came from the interior of Amazonas and belong to riverside isolated populations with low levels of education and poor access to health services. Besides, they have poor knowledge about the disease and seek diagnosis at an advanced stage. This reality was also observed by Chalya et al. (2015) in Tanzania [38] and in Brazil by other groups [33–35,39,40].

PC usually begins with a superficial or ulcerated lesion on the glans and foreskin, but can also spread through the penile shaft and the scrotum [41]. In the current study, the initial lesion was diagnosed in the glans and/or prepuce in 76% of the cases similar to other studies [4,38,42]. Lesions were predominantly of the verruciform type followed by warts. The accumulation of smegma due to poor hygiene followed by probable irritation of the local is a favorable environment for several infections. Chronic inflammatory processes may progress to the development of lesions and if untreated, can lead to neoplasm [7].

In the present study, 70% (23/33) of the patient showed well and moderately differentiated tumors (Grade I/II) and was not associated with HPV infection (*adjusted p-value* = 0.006). Patients with HPV presented higher grade tumor. Histopathology grade is an important prognostic factor. Degree of poorly differentiated cell could indicate a worse prognosis of lesions. Our findings reinforced other studies observations [43,44].

Phimosis, a well-known risk factor for the development of PC, was present in 64% of the patients. 43% were submitted to circumcision during adulthood. Adult circumcision is known to have no protective effect against the development of PC [45]. The univariate logistic regression showed that patients with phimosis had 11 times more chance of overexpression of p16^INK4a [OR = 11 (95%CI 1.1–109.7); p = 0.04]. Among men circumcised in adulthood, phimosis was strongly associated with development of invasive penile cancer, in concordance with the findings of Daling, et al. (2005) [46].

Of note, the presence of HPV in an individual does not mean that the individual will develop cancer. There are many risk factors that contribute to the development or not such as the environment and the genetic background of the individuals as well as the viral clearance capacity of the individual. PC can be HPV-mediated or not. Individuals HPV$^-$ with phimosis and chronic inflammation often develop PC. Genetic and molecular changes associated with HPV$^-$ PC leading to disturbance of the p14$^{ARF}$/MDM2/p53 and/or p16$^{INK4a}$/cyclin D/Rb pathways have been suggested as plausible mechanisms for the development of PC [47]. One study showed that there is silencing of the p16$^{INK4a}$ gene through promoter hypermethylation in 15% of cases and over-expression of the polycomb group (PcG) gene BMI-1, which targets the INK4A/ARF locus, encoding both p16$^{INK4a}$ and p14$^{ARF}$, in 10% of cases. Another study have suggested that the inactivation of p14$^{ARF}$/MDM2/p53 pathway as well as somatic mutation of the p53 gene and over-expression of MDM2 and mutation of p14$^{ARF}$ may lead to the development of PC [48,49].

Emerging interest regarding PC carcinogenesis is the association of oncogenic viruses co-infection. EBV is associated with several malignancies in humans and its involvement in PC is still controversial [50–52]. In this study, the prevalence of EBV was 30% and co-infection with HPV was 29%. Our study differs to the one conducted in Rio de Janeiro where the prevalence of EBV in penile malignancies was 46% and co-infection with HPV was 26% [17]. In cervical cancer, EBV has been suggested as a cofactor that facilitates the integration of the HPV16 genome, contributing to the development of cancer [20,21].

The prevalence of HPV in invasive PC is approximately 45%, ranging from 30% to 75% according to the detection method, the population and type of sample analyzed [11,17,18,40,42,53–55]. High-risk HPV genotype 16 was observed in 61% of HPV$^+$ cases, reinforcing other studies observations [9,56]. HPV 18, the second most common high-risk HPV [11], was not identified in this study. Low incidence of HPV 18 in PC has also been reported in the country [17,40,42,54]. Of note, the prevalence of HPV 18 is also low in the female population of the Amazonas region [57,58]. Interestingly, in one study of the Thailand population, only 1/65 patients with PC had HPV16 but high presence of HPV18 genotype (55%) was detected [59]. This can probably be explained due to geographical distribution of HPV genotypes.

The occurrence of viral co-infection between HPV genotypes 6, 16, 42, 44/55 and EBV was observed. One patient presented EBV/HPV 6 co-infection. Similar findings, EBV/HPV 6 had been described [18] suggesting a probable viral synergism in tumor development due to their similar tropism of epithelial cells [20,21]. The association of EBV and carcinogenesis is still to be demonstrated with EBV genome or virus gene products within the tumor cell population [60]. HPV 6 is classified as low oncogenic risk and is related to condyloma [61]. However, it is a prominent feature in infections in cases of PC and has multiple co-infections with high-risk types [11,62,63].

The overexpression of p16$^{INK4a}$, a surrogate sensitive marker of HPV in PC [16], still remains to correlate with prognostics [12]. Overexpression of P16$^{INK4a}$ was observed in 12 cases (12/26, 46%). 66% (8/12) were from patients infected with HPV 16, reinforcing the role played by HPV16 in the oncogenic process.

Four HPV negative patients showed overexpression of p16$^{INK4a}$ (4/12, 33%). Bleeker et al. (2009) in a systematic review, assumes that PC would be related to a pathway mediated by HPV infection and another due to different epigenetic changes in the absence of HPV and related to chronic inflammation [47]. Understanding the molecular processes involved in the onset and progression of the disease is fundamental for the prevention and treatment of this mutilating.

In this series of cases, the mortality rate during the study was 38%. Several studies tried to identify prognostic factors to manage the selection of patients at high risk for metastases in PC [64–68]. HPV infections as well as co-infection with EBV and p16$^{INK4a}$ positivity were not predictive of survival of the patients with PC. Lymph node involvement is related to poor prognosis, with a 5-year survival of less than 40% [69,70]. In this study, only 39% of the patients with PC had lymphadenectomy.

Limitations of the study were the difficulties inherent to the non-recording of clinical and histopathological data in medical records. Many biopsy samples fixed in paraffin blocks were missing for the IHC assay for the qualitative detection of the p16$^{INK4a}$, reducing the sample size. In the state of Amazonas, usually patients seek care at an advanced stage of the disease and as soon as they complete their surgical treatment (partial or total penectomy) they return to their city and there is no follow-up.

HPV vaccination has been shown to be effective for HPV-related cancers and inclusion of young in the immunization programs against HPV is well established [71,72]. In the Amazonas, the vaccination program for girls started in 2013 with the quadrivalent vaccine that protects against genotypes 6, 11, 16 and 18. In Brazil, the Ministry of Health only included young males (between 12 and 13 years old) in the vaccination HPV program in 2017. In the future, we expect that this action may reduce the incidence of PC and other HPV-related.

## Conclusions

In summary, our results show that patients with HPV$^+$ PC have in general low grade tumors. Overexpression of p16$^{INK4a}$ was correlated to the detection of HPV 16 DNA, reinforcing that it can be used as a marker to high-risk HPV genotype 16 infection as found in oropharyngeal cancers. EBV infection was observed in one-third of the patients with PC and the co-infection with HPV in a quarter. The knowledge of the etiology of penile cancer is far from definite. The individual role or synergisms of the known oncogenic viruses such as HPV and EBV at the onset of carcinogenic events are still not well defined. However, our data show the profile of these viral oncogenic infections and the reality of this neoplasm in individuals from the Brazilian Amazon.

## Supporting information

**S1 Protocol. HPV detection and genotyping method and EBV detection.**
(DOCX)

**S2 Protocol. Immunohistochemical for p16$^{INK4a}$ protein.**
(DOCX)

**S1 Fig. Patterns of p16 expression in penile carcinomas.** Microarray tissue block immunohistochemistry for p16$^{INK4a}$ (from left to right): a. absence; b. strong and diffuse cytoplasmic staining; c. Moderate and focal cytoplasmic staining; d. weak and focal cytoplasmic staining; e. absence of staining; f. strong and diffuse cytoplasmic staining.
(TIF)

**S1 Table. Distribution of HPV, EBV status and p16$^{INK4a}$ overexpression among the patients deceased and survived with penile cancer at Amazon—Brazil.** EBV: Epstein-Barr virus; HPV: Human papillomavirus, n: Absolute frequency; +:positive; -:negative.
(DOCX)

**S1 File. Data collection instrument—Penis cancer.**
(DOCX)

## Acknowledgments

We thank the Fundação Centro de Controle de Oncologia do Estado do Amazonas (FCE-CON), Fundação de Medicina Tropical Doutor Heitor Vieira Dourado (FMT-HVD) and A.C. Camargo Cancer Center for the infrastructural support. The authors gratefully acknowledge Dr. Fernando Augusto Soares and Dra. Stephania Martins Bezerra (A.C. Camargo Cancer Center) who performed the Immunohistochemistry for p16INK4a, Dr. Marcel Heibel for assistance during the inclusion of patients in the study and Dra. Ana Carolina Soares de Oliveira who performed the HPV Genotyping—PapilloCheck® HPV-Screening. The authors would also like to thank the patients that participated in this study.

## Author Contributions

**Conceptualization:** Valquíria do Carmo Alves Martins, José Eduardo Levi, Kátia Luz Torres.

**Data curation:** Valquíria do Carmo Alves Martins, Giuseppe Figliuolo, Heidy Halanna de Melo Farah Rondon, Paloma Menezes de Souza, Felipe Luz Torres Silva, Guilherme Luz Torres Silva.

**Formal analysis:** Daniel Barros de Castro, Monique Freire Santana.

**Funding acquisition:** Kátia Luz Torres.

**Investigation:** Valquíria do Carmo Alves Martins, Isabela Werneck Cunha, Giuseppe Figliuolo, Heidy Halanna de Melo Farah Rondon, Paloma Menezes de Souza, Felipe Luz Torres Silva, Guilherme Luz Torres Silva, Michele de Souza Bastos, Monique Freire Santana, Kátia Luz Torres.

**Methodology:** Valquíria do Carmo Alves Martins, Isabela Werneck Cunha, Heidy Halanna de Melo Farah Rondon, Michele de Souza Bastos, Rajendranath Ramasawmy, Kátia Luz Torres.

**Project administration:** Valquíria do Carmo Alves Martins, José Eduardo Levi, Kátia Luz Torres.

**Resources:** Kátia Luz Torres.

**Supervision:** Valquíria do Carmo Alves Martins, Kátia Luz Torres.

**Validation:** Isabela Werneck Cunha, José Eduardo Levi.

**Writing – original draft:** Valquíria do Carmo Alves Martins, Isabela Werneck Cunha, Rajendranath Ramasawmy, José Eduardo Levi, Kátia Luz Torres.

**Writing – review & editing:** Valquíria do Carmo Alves Martins, Isabela Werneck Cunha, Rajendranath Ramasawmy, José Eduardo Levi, Kátia Luz Torres.

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
