## [Decision Letter · Decision Letter 0]

14 Jan 2020

PONE-D-19-35033

HIGH PREVALENCE OF HPV WITH OVEREXPRESSION OF p16 INK4A PROTEIN AND CO-INFECTION WITH EBV IN PENILE CANCER – A SERIES OF CASES FROM BRAZIL AMAZON

PLOS ONE

Dear Dr. Martins,

Thank you for submitting your manuscript to PLOS ONE. After careful consideration, we feel that it has merit but does not fully meet PLOS ONE’s publication criteria as it currently stands. Therefore, we invite you to submit a revised version of the manuscript that addresses the points raised during the review process.

We would appreciate receiving your revised manuscript by Feb 28 2020 11:59PM. To enhance the reproducibility of your results, we recommend that if applicable you deposit your laboratory protocols in protocols.io, where a protocol can be assigned its own identifier (DOI) such that it can be cited independently in the future. For instructions see: http://journals.plos.org/plosone/s/submission-guidelines#loc-laboratory-protocols

We look forward to receiving your revised manuscript.

Kind regards,

Marc O. Siegel, MD

Academic Editor

PLOS ONE

Journal Requirements:

2. Please include additional information regarding the survey or questionnaire used in the study and ensure that you have provided sufficient details that others could replicate the analyses.

For instance, if you developed a questionnaire as part of this study and it is not under a copyright more restrictive than CC-BY, please include a copy, in both the original language and English, as Supporting Information.

3. To comply with PLOS ONE submission guidelines, in your Methods section, please provide additional information regarding your statistical analyses.

For more information on PLOS ONE's expectations for statistical reporting, please see https://journals.plos.org/plosone/s/submission-guidelines.#loc-statistical-reporting.

4. We ask that you please remove citations for unavailable and unpublished work, including manuscripts that have been submitted but not yet accepted (e.g., “unpublished work,” “data not shown”). Instead, include those data as supplementary material or deposit the data in a publicly available database.

5. In your Methods section, you have indicated that the performed immunohistochemical analysis. At this time, we ask that you please provide the representative microscopy images of the immunohistochemical stainings as a separate Figure in your manuscript.

6. Thank you for stating the following in the Financial Disclosure section:

'The author(s) received specific funding for this work.

1. Kátia Luz Torres - PROGRAMA UNIVERSAL AMAZONAS N.021/2011-FUNDAÇÃO DE AMPARO À PESQUISA DO ESTADO DO AMAZONAS – FAPEAM

2. Kátia Luz Torres - PROGRAMA PAPAC N.020/2013 - FUNDAÇÃO DE AMPARO À PESQUISA DO ESTADO DO AMAZONAS – FAPEAM'

We note that one or more of the authors are employed by a commercial company:

Genomic Health Surveillance Network and Rede D’OR- São Luiz.

Reviewers' comments:

Reviewer's Responses to Questions

**Comments to the Author**

1. Is the manuscript technically sound, and do the data support the conclusions?

Reviewer #1: Yes

Reviewer #2: Partly

2. Has the statistical analysis been performed appropriately and rigorously? 

Reviewer #1: Yes

Reviewer #2: Yes

3. Have the authors made all data underlying the findings in their manuscript fully available?

Reviewer #1: Yes

Reviewer #2: No

4. Is the manuscript presented in an intelligible fashion and written in standard English?

Reviewer #1: Yes

Reviewer #2: No

5. Review Comments to the Author

Reviewer #1: The authors present tumor tissue analyzes from 47 patients with penile cancer. Samples were analyzed for the presence of HPV and EBV DNA by PCR. In addition p16 INK4a expression was evaluated by immunohistochemistry. The results roughly agree with those in the world literature. In the second sentence of the introduction the authors state: In Brazil, PC accounts for approximately 2.1% of all tumors in men being the highest incidence reported in the world (2.9-6.8 cases per 100,000 men-years). In the cited article, Brazil appears in the first place in the ranking of Latin America being surpassed in the world by Romania and Uganda. The authors presented a survival curve showing a better prognosis for Ebv- patients when compared to Ebv +. I would like they comment on why this result.

Reviewer #2: The present study is about penile cancer and oncogenic viruses HPV and EBV and their relationship with clinical and pathological characteristics. The work is important because penile carcinoma is an uncommon neoplasm, although in some regions of the world, such as Brazil, its incidence is higher, and because of the importance of exploring another possible etiological or risk factor, that is, EBV. HPV in penile cancer has been recognized in almost 50% of the cases. In contrast, the presence and biological and clinical relevance of EBV in penile cancer have not been sufficiently studied. However, it is necessary to review the scope of the findings and analyze the concomitant presence of both viruses and the clinical and pathological characteristics of the patients, in order to define the possible relationship of EBV in penile cancer. In this sense, it is necessary to consider the following observations.

Title

“HIGH PREVALENCE OF HPV WITH OVEREXPRESSION OF p16 INK4A PROTEIN AND CO-INFECTION WITH EBV IN PENILE CANCER – A SERIES OF CASES FROM BRAZIL AMAZON”. The title suggests that the authors found a higher prevalence of HPV than the one reported worldwide and that there is a concomitant presence of EBV infection in HPV positive cases. However, the data obtained does not reflect this. I suggest that the title should be modified to reflect in the best way the scope of the study and the results obtained.

Abstract

Line 27. It reads “Epstein Baar”; it should read “Epstein-Barr”.

Line 57. It reads “Con-infection”; it should read “co-infection”.

The conclusion states that “p16 INK4a positivity presented a high correlation to HPV 16 DNA detection, reinforcing its use as a surrogate marker for HPV-driven cancers.” This is not a novel finding. Thus, it is necessary to rewrite the conclusion in the abstract and the text after analyzing the group of concomitant HPV and EBV infection and EBV alone.

Introduction

Line 56-58. It is necessary to update the references about epidemiological data of penile carcinoma, since the only reference is from 2010, and there are recent works on this issue.

Line 69-70. Concerning HPV prevalence in penile carcinoma, it is necessary to complement and or compare the most recent data worldwide (Bruni 2017; Alemany 2016; Olesen 2019).

Line 80-86. It is important to describe the oncogenic role of EBV in cancer, as currently known, and the cell types affected by such viruses. The relationship between EBV and HPV in carcinogenesis is described in various papers; one of them is a review by Guidry and Scott (doi:10.1016/j.virusres.2016.11.002.). It is necessary to expand the references about the oncogenic role of EBV in epithelial carcinogenesis.

Line 87-88. It is necessary to describe in a better way the rationale or the relevance of the work, emphasizing the study of EBV and HPV, and the few studies specifically in penile carcinoma.

Materials and Methods

Line 112-113. Specify the size of the b-globin product.

Line 126. What is the reason to do specific E7 HPV 16 and 18? The results and discussion about this detection are not mentioned. Besides, the genotyping by papillocheck test was used. The rationale to detect E7 is not indicated.

Line 171. It is necessary to indicate why only 21/47 samples were analyzed.

Results

Tables 1 and 2 could be merged.

Line 210-211. In patients with EBV and HPV co-infection, the relationship with p16 overexpression should be described.

Table 3. Include the information about the expression of p16.

Line 215-216. It would be important to evaluate the presence of HPV newly (with a different method, if possible) in the positive cases of p16, which were initially identified as negative for HPV DNA.

Line 219-220. Smoking and HPV presence are relatively excluding factors, at least in HNCSS. The relationship between overexpression of p16 and smoking was identified; however, it was not so with the presence of HPV. The analysis was made with all genotypes identified (low and high risk). What is the relationship between smoking and high-risk HPV presence?

Line 223-228. It is necessary to describe the survival analysis of the group with co-infection of HPV and EBV since the analysis was only made individually, and an important issue (since the title of the study) is the co-infection of EBV and HPV. The result should be discussed.

Table 4. This table could be simplified by removing the percentages (leaving the frequencies), and adding the EBV presence in the header row and respective analysis.

Discussion

Line 251. The metastasis analysis should be extended. This issue has been addressed in head and neck carcinomas HPV negative versus positive, and also in nasopharyngeal carcinomas EBV positive, as well as prognosis factors associated.

Line 252-253. The result about HPV in young men contrasts with a recent article (doi.org/10.1186/s12879-019-4696-6), reporting that at least 50% of men under 45 years have HPV. It is important to extend the discussion about the characteristics of penile cancer in young men and contrast the results with other reports.

Line 258. Change the capital letters in the author's name cited.

Line 268. According to the data in Table 4, tumor grade III is associated with HPV presence. Improve the description of this finding and mention the concordance with previous works.

Line 275-276. How could you explain the relationship between phimosis and overexpression of p16?

Line 279-285. The relationship between EBV and HPV should be expanded. Although the relation between EBV and HPV in penile cancer is incipient, the relationship is described in another HPV-related cancer. Several articles have described the association between expression of p16 and EBV in other neoplasms (Doi: 10.1002/hed.24258. doi.org/10.1016/j.ijrobp.2017.06.1473. doi.org/10.3389/fonc.2018.00113), and this is not described in the discussion.

Line 289-290. The absence of HPV 18 in penile carcinoma is consistent with other works. (doi.org/10.1186/s12879-019-4696-6), and in contrast with the findings in other regions (DOI: 10.1002/jmv.20703).

Line 295. Describe more broadly the possible mechanisms for the synergistic effect of HPV 6 and EBV.

Line 302-303. What other genotypes (different to HPV 16) were present in the p16 positives cases?

Line 304. Change the capital letters of the author's name cited.

Line 313, Figure 2. The correlation between p16 overexpression and survival has been addressed in other neoplasms. The authors should review the impact of the clinical stage on survival. Did you consider the local and advanced stages for survival analysis?

Finally, the authors should discuss the limitations of the study. For example, the number of p16 samples analyzed, if the size (3-5 mm) of samples analyzed was adequate for the detection of the viruses, p16, and pathological characteristics.

One issue not mentioned is the presence of HPV without relation to penile carcinogenesis. Therefore, it is important to describe the p16 overexpression as a subrogate biomarker of carcinogenesis induced by high-risk HPV. The analysis of EBV dependent and independent of HPV should be extended and, consequently, the discussion.

6. PLOS authors have the option to publish the peer review history of their article (what does this mean?). If published, this will include your full peer review and any attached files.

Reviewer #1: No

Reviewer #2: No

---

## [Author Response · Author response to Decision Letter 0]

19 Mar 2020

Reply to reviewers

Reviewer #1:

The authors present tumor tissue analyzes from 47 patients with penile cancer. Samples were analyzed for the presence of HPV and EBV DNA by PCR. In addition, p16 INK4a expression was evaluated by immunohistochemistry. The results roughly agree with those in the world literature. In the second sentence of the introduction the authors state: In Brazil, PC accounts for approximately 2.1% of all tumors in men being the highest incidence reported in the world (2.9-6.8 cases per 100,000 men-years). In the cited article, Brazil appears in the first place in the ranking of Latin America being surpassed in the world by Romania and Uganda. 

We agree and have corrected in the texts. (Line 57-58)

The authors presented a survival curve showing a better prognosis for Ebv- patients when compared to Ebv +. I would like they comment on why this result.

We agree with the reviewer that there is a trend due to the sudden drop of the survival curve. However, the comparison showed no difference (Plog rank = 0.106).

Reply to reviewers

Reviewer #2: 

The present study is about penile cancer and oncogenic viruses HPV and EBV and their relationship with clinical and pathological characteristics. The work is important because penile carcinoma is an uncommon neoplasm, although in some regions of the world, such as Brazil, its incidence is higher, and because of the importance of exploring another possible etiological or risk factor, that is, EBV. HPV in penile cancer has been recognized in almost 50% of the cases. In contrast, the presence and biological and clinical relevance of EBV in penile cancer have not been sufficiently studied. However, it is necessary to review the scope of the findings and analyze the concomitant presence of both viruses and the clinical and pathological characteristics of the patients, in order to define the possible relationship of EBV in penile cancer. In this sense, it is necessary to consider the following observations.

Title

“HIGH PREVALENCE OF HPV WITH OVEREXPRESSION OF p16 INK4A PROTEIN AND CO-INFECTION WITH EBV IN PENILE CANCER – A SERIES OF CASES FROM BRAZIL AMAZON”. The title suggests that the authors found a higher prevalence of HPV than the one reported worldwide and that there is a concomitant presence of EBV infection in HPV positive cases. However, the data obtained does not reflect this. I suggest that the title should be modified to reflect in the best way the scope of the study and the results obtained.

We agree and have rewritten the title.

PRESENCE OF HPV WITH OVEREXPRESSION OF p16 INK4A PROTEIN AND EBV INFECTION IN PENILE CANCER – A SERIES OF CASES FROM BRAZIL AMAZON 

Abstract

Line 27. It reads “Epstein Baar”; it should read “Epstein-Barr”.

We have corrected in the texts. (Line 29)

Line 57. It reads “Con-infection”; it should read “co-infection”.

We have corrected in the texts. (Line 51-52)

The conclusion states that “p16 INK4a positivity presented a high correlation to HPV 16 DNA detection, reinforcing its use as a surrogate marker for HPV-driven cancers.” This is not a novel finding. Thus, it is necessary to rewrite the conclusion in the abstract and the text after analyzing the group of concomitant HPV and EBV infection and EBV alone.

We totally agree and have brought the necessary changes. (Line 51-52)

Introduction

Line 56-58. It is necessary to update the references about epidemiological data of penile carcinoma, since the only reference is from 2010, and there are recent works on this issue.

We have brought the necessary changes. (Line 55-57)

Line 69-70. Concerning HPV prevalence in penile carcinoma, it is necessary to complement and or compare the most recent data worldwide (Bruni 2017; Alemany 2016; Olesen 2019). 

We have brought the necessary changes. (Line 69-71)

Line 80-86. It is important to describe the oncogenic role of EBV in cancer, as currently known, and the cell types affected by such viruses. The relationship between EBV and HPV in carcinogenesis is described in various papers; one of them is a review by Guidry and Scott (doi:10.1016/j.virusres.2016.11.002.). It is necessary to expand the references about the oncogenic role of EBV in epithelial carcinogenesis.

We have corrected accordingly. (Line 81-88)

Line 87-88. It is necessary to describe in a better way the rationale or the relevance of the work, emphasizing the study of EBV and HPV, and the few studies specifically in penile carcinoma.

We totally agree and have brought the necessary changes. (Line 89-90)

Materials and Methods

Line 112-113. Specify the size of the b-globin product.

We have provided in the text. (Line 115-117)

Line 126. What is the reason to do specific E7 HPV 16 and 18? The results and discussion about this detection are not mentioned. Besides, the genotyping by PapilloCheck test was used. The rationale to detect E7 is not indicated.

Reply

As PapilloCheck is a very expensive test, we have chosen to primarily screen the presence of HPV followed by typing specifically for HPV16 and HPV18. All of the samples that were negative for HPV16 and HPV18 were screen by PapilloCheck test to reduce the cost of typing.

Line 171. It is necessary to indicate why only 21/47 samples were analyzed.

Reply

We believe that there is misunderstanding here. We did already explain in the text. 21 samples were not available for pre-analytical analysis due to poor quality keeping of the samples.

Results

Tables 1 and 2 could be merged.

We do agree but merging both tables 1 and 2 will be heavy and confusing.

Line 210-211. In patients with EBV and HPV co-infection, the relationship with p16 overexpression should be described.

We have provided in the text. (Line 223-224)

Table 3. Include the information about the expression of p16.

Reply

As we did for only 26 samples, we did not include in the Table 3 to avoid confusion, but we describe in text.

Line 215-216. It would be important to evaluate the presence of HPV newly (with a different method, if possible) in the positive cases of p16, which were initially identified as negative for HPV DNA.

Reply 

The four samples that were positive for p16 and negative for HPV were again tested for the presence of HPV16 and HPV18 by real-time PCR and continued to be negative. 

Line 219-220. Smoking and HPV presence are relatively excluding factors, at least in HNCSS. The relationship between overexpression of p16 and smoking was identified; however, it was not so with the presence of HPV. The analysis was made with all genotypes identified (low and high risk). What is the relationship between smoking and high-risk HPV presence?

Reply

We have rewritten in the result section to avoid confusion. There was no relationship between smoking and high-risk HPV presence. As it can be seen that the driving force for the presence of p16 is mostly dependent on the presence of HPV and independent of the smoking status. (Line 232-238)

Line 223-228. It is necessary to describe the survival analysis of the group with co-infection of HPV and EBV since the analysis was only made individually, and an important issue (since the title of the study) is the co-infection of EBV and HPV. The result should be discussed.

Thank you for pointing this lack. We have performed the analysis and is shown in Figure 2D. The comparison showed no difference P log rank=0.318. We also included a S1 Table showing the infection status of the deceased and survived patients. (Line 240-241)

Table 4. This table could be simplified by removing the percentages (leaving the frequencies) and adding the EBV presence in the header row and respective analysis.

We have brought the necessary changes.

Discussion

Line 251. The metastasis analysis should be extended. This issue has been addressed in head and neck carcinomas HPV negative versus positive, and also in nasopharyngeal carcinomas EBV positive, as well as prognosis factors associated.

Reply 

As our study is only focused on penile cancer, we did not compare with other types of cancers as we believe that the developing mechanisms could be different. For this reason, we did not extend the discussion.

Line 252-253. The result about HPV in young men contrasts with a recent article (doi.org/10.1186/s12879-019-4696-6), reporting that at least 50% of men under 45 years have HPV. It is important to extend the discussion about the characteristics of penile cancer in young men and contrast the results with other reports.

Reply

We agree to disagree because of the sample size which makes it difficult to compare. In the study stated, there were only 8 patients under 45 years old and four of them were positive for HPV. In our study we had 13 patients under 45 years old and five (38%) were positive for HPV. 

Line 258. Change the capital letters in the author's name cited.

We have corrected in the texts. (Line 259)

Line 268. According to the data in Table 4, tumor grade III is associated with HPV presence. Improve the description of this finding and mention the concordance with previous works.

We have added one more reference. (Line 292)

Line 275-276. How could you explain the relationship between phimosis and overexpression of p16?

We have provided in the discussion. (Line 300-312)

Line 279-285. The relationship between EBV and HPV should be expanded. Although the relation between EBV and HPV in penile cancer is incipient, the relationship is described in another HPV-related cancer. Several articles have described the association between expression of p16 and EBV in other neoplasms (Doi: 10.1002/hed.24258. doi.org/10.1016/j.ijrobp.2017.06.1473. doi.org/10.3389/fonc.2018.00113), and this is not described in the discussion.

Reply 

As cited previously (Answer Line 252). We have focus mainly on penile cancer. Considering that in other studies with penile cancer the HPV and EBV status were not predictive of outcome and we believe that the developing mechanisms could be different for another cancer.

Line 289-290. The absence of HPV 18 in penile carcinoma is consistent with other works. (doi.org/10.1186/s12879-019-4696-6), and in contrast with the findings in other regions (DOI: 10.1002/jmv.20703).

We have included in the text. (Line 326-329)

Line 295. Describe more broadly the possible mechanisms for the synergistic effect of HPV 6 and EBV.

We included in the text. (Line 333)

Line 302-303. What other genotypes (different to HPV 16) were present in the p16 positives cases?

We have described in the result section. (Line 229-229)

Line 304. Change the capital letters of the author's name cited.

We have corrected in the texts. (Line 342)

Line 313, Figure 2. The correlation between p16 overexpression and survival has been addressed in other neoplasms. The authors should review the impact of the clinical stage on survival. Did you consider the local and advanced stages for survival analysis?

Reply

We have included in the text. However, we did stratify the patients according to the local and advanced stages as our sample size is small. And stratification will further reduce the sample and will not have any power to detect any difference according to our statistician. (Line 350-351)

Finally, the authors should discuss the limitations of the study. For example, the number of p16 samples analyzed, if the size (3-5 mm) of samples analyzed was adequate for the detection of the viruses, p16, and pathological characteristics.

We have described in the discussion section. And we also described in the materials and methods section and Supporting Information (S2 PROTOCOL - Immunohistochemistry for p16INK4a).

One issue not mentioned is the presence of HPV without relation to penile carcinogenesis. Therefore, it is important to describe the p16 overexpression as a surrogate biomarker of carcinogenesis induced by high-risk HPV. The analysis of EBV dependent and independent of HPV should be extended and, consequently, the discussion.

We have described in the discussion section.

Sincerely yours, 

Yours faithfully,

Valquiria do Carmo Alves Martins

---

## [Decision Letter · Decision Letter 1]

16 Apr 2020

PRESENCE OF HPV WITH OVEREXPRESSION OF p16 INK4A PROTEIN AND EBV INFECTION IN PENILE CANCER – A SERIES OF CASES FROM BRAZIL AMAZON

PONE-D-19-35033R1

Dear Dr. Martins,

We are pleased to inform you that your manuscript has been judged scientifically suitable for publication and will be formally accepted for publication once it complies with all outstanding technical requirements.

With kind regards,

Marc O. Siegel, MD

Academic Editor

PLOS ONE

Additional Editor Comments (optional):

Reviewers' comments:

Reviewer's Responses to Questions

**Comments to the Author**

1. If the authors have adequately addressed your comments raised in a previous round of review and you feel that this manuscript is now acceptable for publication, you may indicate that here to bypass the “Comments to the Author” section, enter your conflict of interest statement in the “Confidential to Editor” section, and submit your "Accept" recommendation.

Reviewer #1: All comments have been addressed

Reviewer #2: All comments have been addressed

2. Is the manuscript technically sound, and do the data support the conclusions?

Reviewer #1: Yes

Reviewer #2: Yes

3. Has the statistical analysis been performed appropriately and rigorously? 

Reviewer #1: Yes

Reviewer #2: Yes

4. Have the authors made all data underlying the findings in their manuscript fully available?

Reviewer #1: Yes

Reviewer #2: Yes

5. Is the manuscript presented in an intelligible fashion and written in standard English?

Reviewer #1: Yes

Reviewer #2: Yes

6. Review Comments to the Author

Reviewer #1: The authors made the changes that I requested. The manuscript deserves to be published due to the rarity of the tumor studied. In addition, it is important to have a sample of the population of Amazonas in northern Brazil. Mainly because it includes minority groups such as indigenous people.

Reviewer #2: The authors have corrected and incorporated most of the previous suggestions and they have answered the questions. However, it is still necessary to correct some edition errors.

Line 29. It reads “Epstein Baar”; it should read “Epstein-Barr”

Lines 251-259 and page 13. The authors should check and make sure that the section and order of the footnotes of the figures are correct and complete.

7. PLOS authors have the option to publish the peer review history of their article (what does this mean?). If published, this will include your full peer review and any attached files.

Reviewer #1: No

Reviewer #2: No

---

## [Editor Report · Acceptance letter]

24 Apr 2020

PONE-D-19-35033R1 

PRESENCE OF HPV WITH OVEREXPRESSION OF p16 INK4A PROTEIN AND EBV INFECTION IN PENILE CANCER – A SERIES OF CASES FROM BRAZIL AMAZON 

Dear Dr. Martins:

I am pleased to inform you that your manuscript has been deemed suitable for publication in PLOS ONE. Congratulations! Your manuscript is now with our production department. 

With kind regards,

on behalf of

Dr. Marc O. Siegel 

Academic Editor

PLOS ONE